# The Tribological and Mechanical Properties of PI/PAI/EP Polymer Coating under Oil Lubrication, Seawater Corrosion and Dry Sliding Wear

**DOI:** 10.3390/polym15061507

**Published:** 2023-03-17

**Authors:** Shijie Yu, Jun Cao, Shuxin Li, Haibo Huang, Xiaojie Li

**Affiliations:** 1School of Mechanical Engineering and Mechanics, Ningbo University, Ningbo 315211, China; 2School of Mechanical and Electrical Engineering, Wenzhou University, Wenzhou 325035, China; 3School of Materials, Southwest Jiaotong University, Chengdu 610031, China

**Keywords:** polymer coating, Ce_2_O_3_, tribology, oil lubrication, dry sliding wear, seawater corrosion

## Abstract

To investigate the tribological performance of a copper alloy engine bearing under oil lubrication, seawater corrosion and dry sliding wear, three different PI/PAI/EP coatings consisting of 1.5 wt% Ce_2_O_3_, 2 wt% Ce_2_O_3_, 2.5 wt% Ce_2_O_3_ were designed, respectively. These designed coatings were prepared on the surface of CuPb22Sn2.5 copper alloy using a liquid spraying process. The tribological properties of these coatings under different working conditions were tested. The results show that the hardness of the coating decreases gradually with the addition of Ce_2_O_3_, and the agglomeration of Ce_2_O_3_ is the main reason for the decrease of hardness. The wear amount of the coating increases first and then decreases with the increase of Ce_2_O_3_ content under dry sliding wear. The wear mechanism is abrasive wear under the condition of seawater. The wear resistance of the coating decreases with the increase of Ce_2_O_3_ content. The wear resistance of the coating with 1.5 wt% Ce_2_O_3_ is the best under-seawater corrosion. Although Ce_2_O_3_ has corrosion resistance, the coating of 2.5 wt% Ce_2_O_3_ has the worst wear resistance under seawater conditions due to severe wear caused by agglomeration. Under oil lubrication conditions, the frictional coefficient of the coating is stable. The lubricating oil film has a good lubrication and protection effect.

## 1. Introduction

Copper bearings are key parts of aeroplane, ship, and automobile engines [1,2,3]. At the engine start and stop stages, the bearings are in a state of boundary lubrication and even dry sliding wear [4,5]. Copper alloys have excellent tribological properties under oil lubrication. Yin et al. [6] showed that the coefficient of friction (CoF) between tin bronze and steel at a load of 30 N and a speed of 16 mm/s was 0.11. Zhu et al. [7] reported that the CoF of a CuNiSn alloy under oil lubrication was 0.11 at a load of 1 N and a sliding speed of 20 mm/s. Ajay et al. [8] confirmed that the CoF of a bronze alloy under oil lubrication was 0.065 at a load of 35 N and a sliding speed of 100 mm/s. At some times, the engine bearings will face extreme working conditions. For example, an engine room of a ship will be hit by shells and affected by air shock waves during a war, causing the bearings to be in a state of dry sliding wear and seawater infiltration. The tribological properties and work performances of bearing alloys will seriously decline under dry friction sliding wear and seawater corrosion. Fan et al. [9] reported that the CoF of a QSn7-0.2 copper alloy under dry sliding wear was 0.67 at a contact load of 72 N and a sliding speed of 360 mm/s. Nai et al. [10] confirmed that the CoF of a Cu-1%Cr alloy was 0.316 at a load of 30 N and a sliding speed of 1500 mm/s. Yu et al. [11] showed a CoF of 0.43 for a copper alloy with GGr15 steel balls at a load of 5 N and a speed of 12 mm/s. The above studies show that the CoFs of copper alloys rise sharply under dry sliding wear, which will easily lead to bearings damage [12]. Under seawater conditions, the tribological performance of copper alloy bearings is affected by both water film lubrication and seawater corrosion. Wang et al. [13] confirmed that the CoF of a C72700 copper alloy with a frictional pair of Si_3_N_4_ balls in a 3.5 wt% NaCl solution at a speed of 136 mm/s and a load of 1 N was 0.55. Cui et al. [14] showed that the CoF of Cu-6Sn-6Zn -3Pb copper alloy at a load of 50 N in seawater was 0.21. Drach et al. [15] reported a corrosion rate of 13.9 µm/year for copper in natural seawater. Ida et al. [16] studied the corrosion rate of a C70600 copper alloy in different NaCl solutions at room temperature and the effect of corrosion on the copper material. The results showed that the corrosion rate of the copper alloy in a 0.63 mol-L^−1^ NaCl solution was the largest, at 1.89 mpy. According to the above studies, seawater has a corrosive effect on copper alloy bearings, and the mechanical properties of the bearings are decreased, leading to a reduction in wear resistance.

The aforementioned studies demonstrate that a bearing alloy’s corrosion resistance and self-lubricating qualities are the primary determinants of the bearing’s tribological resistance under difficult operating conditions. Surface technology can be used to enhance the bearing performance under difficult conditions because the wear and corrosion resistance of the material depends on its surface properties [17]. The mainstream surface treatments, such as Physical Vapor Deposition (PVD), electroplating, and thermal spraying, have the disadvantages of low production efficiency and high cost, environmental pollution and high oxidation, respectively [18,19]. They are not suitable for the plain engine bearing industry. A polymer coating is applied to the surface of an engine bearing. Xi [20] prepared a solid lubricant coating by in situ synthesis of cuprous sulphide (Cu_2_S) nanoparticles in polyamide-imide (PAI) and polytetrafluoroethylene (PTFE), and tests showed that the coating had a CoF of 0.071 under dry sliding wear at a phase load of 5 N. Song et al. [21] prepared TiN and TiN-MoS_2_/PTFE coatings, respectively, and the tests showed that the CoF of the TiN coating was 0.29 and 0.095, respectively, at a load of 80 N and a speed of 8 mm/s. Duan et al. [22] prepared a Triton X-100-modified fluorinated graphene and ZIF-8 composite epoxy coating, and the tests showed that the CoF of the coating was 0.15 under dry sliding wear at a load of 5 N and a sliding speed of 10 mm/s. The polymer coating is not only self-lubricating but also corrosion resistant [23,24,25]. Ren et al. [26] prepared an h-BN@PDA epoxy coating and found that the corrosion potential of the composite coating in a 3.5 wt% NaCl solution was 0.1 V. Gu et al. [27] studied the effect of modified graphene oxide (BGO) on the corrosion resistance of epoxy coatings and found that the resistance of the epoxy coating with 0.1 wt% BGO was 1.03 GΩ·cm^2^ in a 3.5 wt% NaCl solution.

The above studies show that the polymer coatings have good self-lubrication and corrosion resistance, but there is less research on marine engine bearing in the specific environment of a sudden shell attack, continuous experience of oil lubrication, seawater, or dry sliding wear. It is not clear whether the above conditions are met at the same time. In this study, three different polymetallic coating materials were designed to prepare a coating on a CuPb22.5Sn2.5 copper alloy. This alloy is widely applied for engine bearing. The tribological properties of these coatings were investigated through wear experiments under oil-lubricated, dry sliding wear, and seawater-lubricated conditions. The tribological mechanisms of the coatings under different operating conditions are described.

## 2. Materials and Methods

### 2.1. Preparation of Coating Materials

The specific composition of the coating materials was as follows: polyimide (PI) and polyamide-imide (PAI), epoxy resin (Ep-44), polytetrafluoroethylene (PTFE), N-N dimethylformamide (DMF), tungsten disulphide (WS_2_), and cerium trioxide (Ce_2_O_3_). The solvent for the preparation of the polymorphic coatings was a mixture of acetone, DMF, and diluent. Ep-44 was the resin binder, which significantly influences the adhesion of the coatings. PI and PAI have excellent mechanical properties and can significantly enhance the wear resistance of the coatings after intermixing and curing with the resin [28,29,30]. PTFE has excellent lubricity and corrosion resistance and can be used as the lubricating phase to reduce the wear of the coatings [31,32]. WS_2_ crystals have a special interlayer crystalline structure, and the layers are connected by van der Waals forces, which have excellent lubricating properties [33,34]. The Ce_2_O_3_ particles can be used as a reinforcing phase for wear and corrosion resistance [35,36].

The polymer coating was prepared in the following steps. First, we configured the solvent mixture of DMF, acetone, and diluent. Second, we put Ep-44, PI, and PAI into the ball mill tank with the solvent, which was ball milled in a KE-0.4 L planetary ball mill for 16 h. Then, we added WS_2_, PTFE, and Ce_2_O_3_ to the ball mill tank, and ball milled the mixture for 16 h. Finally, we added the additives into the ball mill tank and continued to ball mill for 24 h. The ball mill rotation speed was 300 r/min. The anti-corrosion properties, mechanical performance, and tribological properties of coatings vary with the different content of Ce_2_O_3_. The wear resistance of coatings with different CeO_2_ contents was investigated by Ding, and the results showed that the coatings possessed the highest hardness and the lowest wear rate at a CeO_2_ content of 1.5 wt% [36]. The effect of Ce content on the performance of TiN coatings was studied by Fan, and the results showed that the highest adhesion was achieved at a Ce content of 1.9% wt% [37]. Based on the above studies, a particular coating concentration of Ce_2_O_3_ at 1.5 wt%, 2 wt% and 2.5 wt% was designed, respectively. The specific compositions are shown in Table 1.

The CuPb22.5Sn2.5 copper alloy was cleaned with alcohol for oil removal before coating deposition. Then, the dry substrates were sandblasted, and the surface roughness Ra was 0.9 ± 0.1 μm. After that, the copper alloys were preheated at 90 °C for 30 min. Then, the prepared materials were sprayed with an Iwata Aneste (RG-3L) spray gun (Shizuoka, Japan). The spray gun pressure was 0.3 MPa, the spraying distance was 230 ± 20 mm, and the spraying angle was 80 ± 5°. Finally, the spray samples were cured at 220 °C for 2 h. The different coatings obtained were named T1.5, T2, and T2.5, as shown in Table 1.

### 2.2. Tribological Properties Characterization

A high-frequency reciprocating friction and wear testing machine (CMS-01, Beijing Chaoyang High-Tech Applied Technology Research Institute Co., Ltd., Beijing, China) was used to evaluate the tribological and wear properties of the coating under dry sliding wear, seawater lubrication and oil lubrication. In this paper, a marine engine was used as an example with a common pressure of 50 MPa in the engine bearing and a rated spindle speed of 3000 r/min. A stainless steel ball with a radius of 6 mm was used in the experiments. The details of the frictional pair are listed in Table 2.

The test load calculated from the Hertzian contact stress was 2 N, and the contact pressure is close to 50 MPa. Reciprocating sliding friction wear was then applied in frictional tests. To accelerate wear failure, a reciprocating sliding friction at a distance of 1 mm and a frequency of 50 Hz was applied. It was simulated for a rotation speed of 3000 r/min of the engine bearing. The tests were performed at room temperature (23 °C) and a relative humidity of 50 ± 5%. The specific test conditions are shown in Figure 1.

The wear profile was characterized by a laser scanning microscope (LSM-900, Zeiss, Jena, Germany), and the wear rate (*K*, unit: mm^3^N^−1^m^−1^) was calculated by the following formula:(1)$$K=\frac{V}{FL},$$
where *V* is the wear volume (mm^3^), *F* is the applied load (N), and *L* is the total sliding distance (m). Under the same test conditions, the above tribological tests were carried out three times and averaged.

### 2.3. Microstructure Characterization and Mechanical Properties

The phase composition of the coating was characterized by XRD (D8 advance, Bruker, Bremen, Germany). The data were collected using Cu targets in online scanning mode. The current was 40 mA, the voltage was 40 kV, the step size was 0.02°, and the scanning range of the diffraction angle (2θ) was 10~90°. The samples were polished under the same working conditions on an F-VD600 polishing machine, and then the same batch of samples was characterized by scanning electron microscopy (SU500, Hitachi, Fuji City, Japan). The abrasive surface of the coating was observed and analyzed by optical microscopy. A nanoindentation instrument (TI Premier, Hysitron, Eden Prairie, MN, USA) was used to test the hardness of the coating. The test load was 1000 μN, the pressure holding time was 2 s, and the mean values of the six data points were randomly tested.

## 3. Results and Discussion

### 3.1. Coating Phase Analysis

The XRD results for the three different coatings are shown in Figure 2, where the characteristic peaks of WS_2_, Ce_2_O_3_, and CeO_2_ appear. The characteristic diffraction peaks of (002) and (104) crystal planes of WS_2_ appear at 14.3° and 44.2°. The characteristic diffraction peaks of the (003) and (004) crystal planes of Ce_2_O_3_ appear at 44.8° and 61.1°. The characteristic diffraction peak of the (111) crystal plane of CeO_2_ appears at 28.5°. There is a characteristic peak for CeO_2_, which is due to the easy oxidation of Ce_2_O_3_ particles. Zhang et al [38]. confirmed that, after the high-temperature oxidation of a Ce-5La alloy under 200 °C for 2 h, the Ce atom on the surface was first oxidized to Ce_2_O_3_, and finally oxidized to the more stable CeO_2_. The high-temperature curing of the coatings was 220 °C in air holding for 2 h, during which Ce_2_O_3_ particles were oxidized to more stable CeO_2_ particles.

### 3.2. Micromorphology of Coatings

The cross-sectional morphology of these composite coatings are shown in Figure 3. Figure 3a–c shows that the coatings with different Ce_2_O_3_ particle contents are tightly bonded to the substrate, no cracks appear, and the coatings are dense and without pore defects. This is attributed to the Ce ion in Ce_2_O_3_, a rare earth element with a special out-of-nuclear electron arrangement, which can form coordination bonds with hydroxyl and hydroxymethyl groups and which plays a catalytic role in the resin curing, leading to an enhanced coating adsorption capacity [39,40]. In contrast, it can be seen in the enlarged Figure 3d–f that, with an increase in Ce_2_O_3_ addition, Ce_2_O_3_ appears agglomerated in the coating. It can be seen at the mark in Figure 3f that the agglomerated Ce_2_O_3_ particles in the T2.5 coating are very big and unevenly distributed. Figure 3d,e shows that the Ce_2_O_3_ particles in T1.5 and T2 exhibit elongated strips. The EDS scans of the different coatings are shown in Figure 4. From Figure 4, it is clear that the distribution of Ce elements in the T1.5 coating is uniform, which indicates that Ce_2_O_3_ particles can be uniformly distributed in the coating. With the addition of Ce_2_O_3_ particles, Ce elements gradually gather. In the T2.5 coating, Ce_2_O_3_ particles appear to be agglomerated, which is consistent with the large particles in Figure 3f.

### 3.3. Hardness and Elastic Modulus Analysis of the Coatings

The hardness and elastic modulus of coatings with different Ce_2_O_3_ contents are shown in Figure 5. Figure 5 indicates that the hardness of the coating gradually decreases with the increase of Ce_2_O_3_ content. The maximum hardness of the coating is 0.224 GPa for a Ce_2_O_3_ content of 1.5 wt%. At a Ce_2_O_3_ content of 2.5 wt%, the hardness of the coating decreases to 0.164 GPa—a reduction of 26.7%. The elastic modulus of the coating first increases and then decreases with increasing Ce_2_O_3_ content. The minimum elastic modulus of the coating is 2.60 GPa for a Ce_2_O_3_ content of 1.5 wt%. The maximum elastic modulus of the coating with Ce_2_O_3_ at 2 wt% is 3.26 GPa, which is 25.3% higher than that of the coating with 1.5 wt% Ce_2_O_3_. Ce_2_O_3_ is a rare earth compound, which has been shown to have a curing catalytic effect on polymeric organics and can significantly enhance the mechanical properties of polymeric compounds [41]. Therefore, the hardness of the T1.5 and T2 coatings is relatively strong. On the other hand, the polymer hardness depends on the size of the ceramic particles. The increase of particles and the agglomeration of coatings will lead to the uniform distribution of Ce elements in T1.5 and T2 coatings, indicating that Ce_2_O_3_ particles in the coating are evenly distributed, and the coatings hardness increases [42]. In the T2.5 coating, the Ce_2_O_3_ particles show significant clustering. The hardness of the coating is significantly reduced due to the increase in particle size.

### 3.4. Tribological Properties of the Coatings under Oil Lubrication

The normal operation of the engine bearing is oil lubrication. The CoFs of different coatings under oil lubrication working conditions are shown in Figure 6. It is clear that the coating enters into stable wear more quickly under oil lubrication working conditions, and the differences in the average CoFs of different coatings are small. With the rise of Ce_2_O_3_ particle content, the average CoF of the coating first decreases and then increases. The average CoFs of T1.5, T2, and T2.5 coatings are 0.072, 0.065, and 0.067, respectively. The average CoF of T1.5 is the largest, with a 10.8% increase compared with that of T2. During the whole wear process, the CoF of the T1.5 coating is the largest, and the CoF of the T2 coating is the smallest at the beginning. It increases significantly to 0.071 at the late wear period over that of T2.5 coating. Figure 7 shows the wear marks of the coatings under oil lubrication conditions, and it can be seen that the wear mechanism of different coatings under oil lubrication is adhesive wear. The spalling caused by fatigue pitting is visible on the wear marks. As the Ce_2_O_3_ content rises, the area of spalling pits on the wear marks decreases and then rises. The serious spalling of the T1.5 coating is due to its high CoF and severe wear. The coating wear resistance was reduced due to the agglomeration of excessive Ce_2_O_3_ particles. In summary, the T2 coating has the best wear performance under oil lubrication conditions.

### 3.5. Tribological Properties of the Coatings under Dry Sliding Wear

Under the influence of airwaves after being struck by a shell, the engine bearing undergoes a dry sliding wear condition. The CoFs of the coatings under dry sliding wear are shown in Figure 8a. As can be seen, the average CoF of the T2 coating is the lowest at 0.116, whereas the average CoF of the T2.5 coating is the highest at 0.127. The average CoF of the T2.5 coating is increased by 9.5% compared with that of the T2 coating. As the addition of Ce_2_O_3_ particles increases, the run-in time of the coating gradually increases. T1.5, T2, and T2.5 enter the stable wear phase at about 300 s, 400 s and 900 s, respectively. On the one hand, as the number of solid particles increases, the numbers of micro-convex peaks on the coated surfaces increase due to the accumulation of solid particles, and more time is required to smooth the surface against abrasions. On the other hand, the PTFE content in the T1.5, T2, and T2.5 coatings gradually decreased and the lubrication capacity of the coatings decreased. Zhang et al. [43] pointed out that overly low PTFE content in the coating will lead to difficulty in forming a PTFE transfer lubrication film, resulting in decreased coating lubrication ability. Thus, the run-in time of T1.5 is the shortest and that of T2.5 is the longest. After entering the stable wear phase, the CoF of the coating stabilizes as the formation of wear debris and the formation of the transfer film reaches dynamic equilibrium. The transfer film formed at the friction interface acts as a lubricant, causing the CoF to decrease and remain stable. The CoF of the T2 coating is the lowest during the stable wear phase. The reason is that the T2 coating has the highest elastic modulus at the same load and the smallest actual contact area, and therefore the smallest CoF. The initial CoF of the T2.5 coating is low because the coating hardness is the lowest, and the required shear force is the smallest. The reason for the maximum CoF of the T2.5 coating during the stable wear phase is that the aggregated Ce_2_O_3_ particles provide a large resistance during the sliding wear process and the minimal PTFE addition leads to insufficient lubrication. After entering the stable wear phase, the CoF of the T2.5 coating tends to decrease slightly at first and then increase abruptly. The decrease in the CoF is because residual debris at the worn interface forms a transfer film, which acts as a lubricant. Subsequently, the stability of the lubricating film is disturbed due to large particles in the T2.5 coating that are shed from the coating and enter the wear interface, resulting in a sudden increase in the CoF and pronounced spalling pits at the wear mark. The wear rates of different coatings under dry sliding wear are shown in Figure 8b. The wear rates of T1.5, T2 and T2.5 are 6.88 × 10^−6^ mm^3^N^−1^m^−1^, 5.74 × 10^−6^ mm^3^N^−1^m^−1^ and 10.29 × 10^−6^ mm^3^N^−1^m^−1^, respectively. The T2 coating has the lowest wear rate. This is due to its superior hardness and having the highest elastic modulus, which result in a small actual contact area, reduced friction, and increased wear resistance during rubbing. The T2.5 coating has the largest wear rate. On the one hand, the addition of excessive agglomeration of Ce_2_O_3_ particles leads to a reduction in the hardness, which in turn leads to a decrease in wear resistance. On the other hand, the lubrication performance of the coating is reduced due to the reduced lubrication effect of the PTFE content. In summary, in a dry frictional environment, the wear resistance of the coating first increases and then decreases with the addition of Ce_2_O_3_ particles.

The coatings’ surfaces after dry sliding wear were characterized by optical microscopy (KH-8700, Hirox, Tokyo, Japan). The wear morphologies of the coatings under different dry frictional conditions are shown in Figure 9. The T1.5, T2, and T2.5 coatings show obvious abrasive wear characteristics. The worn surface of the T1.5 coating has only a few furrows. This is because the T1.5 coating has the highest hardness, and a small amount of Ce_2_O_3_ particles can hardly produce a noticeable furrow. A large number of furrows and spalling pits can be seen on the worn surface of the T2 coating. This is because the spallation of the Ce_2_O_3_ particles in the process of high frequency reciprocal sliding wear leads to drastic wear, hence the apparent furrows on the worn surface. The T2.5 coating is visible as a few plough grooves and flaking pits, with overall smooth abrasion marks. The smoothness of the abrasion surface is due to the formation of a stable transfer film on the dual surface, so the surface is relatively flat. Lin et al. [44] pointed out that the addition of Ce_2_O_3_ can improve the bonding property of a resin-based coating and the tearing resistance of the frictional interface, as well as promote the formation of the frictional interface lubrication film. As a result, the wear marks of the T2.5 coating are smooth. However, due to the minimal hardness and elastic modulus of the T2.5 coating and its weak resistance to wear, the T2.5 coating has the largest wear width and depth and the highest wear rate. In summary, the T2 coating has the best wear resistance and the T2.5 coating has the worst wear resistance under dry sliding wear.

### 3.6. Tribological Properties of the Coatings under Seawater

After the engine room of a ship is attacked by a shell, the engine bearing enters a state of seawater infiltration. Therefore, the tribological properties of the coatings were tested under the conditions of seawater corrosion. The CoFs of different coatings under seawater lubrication are shown in Figure 10a. Compared with dry sliding wear, the average CoFs of all coatings under seawater lubrication increase significantly, at 0.124, 0.128, and 0.159 for T1.5, T2, and T2.5, respectively. They increased by 3.8%, 10.3%, and 25.2%, respectively. Figure 10a shows that the curve fluctuations of the CoFs under seawater lubrication become significantly greater. This is because the Ce_2_O_3_ particles in the coating are prone to dropping off and entering the frictional interface due to the effect of seawater erosion and the scouring effect of high-frequency reciprocating frictional experiments. It results in three-body wear, which damages the water film that act as lubricants and leads to an increase in the CoF. Subsequently, the Ce_2_O_3_ particles are washed out of the frictional interface by seawater, and the CoF decreases. That is the reason for the fluctuation of the CoFs under the seawater. The wear conditions of the different composite coatings are shown in Figure 10b. The wear rates of T1.5, T2 and T2.5 are 1.47 × 10^−5^ mm^3^N^−1^m^−1^, 2.01 × 10^−5^ mm^3^N^−1^m^−1^, and 3.11 × 10^−5^ mm^3^N^−1^m^−1^, respectively. The T1.5 coating has the lowest wear rate due to its high hardness and strong resistance to wear. The wear rate of the T2.5 coating is the greatest, and is increased by 111.6% compared with that of T1.5. The reason is that the hard particles in the T2.5 coating are aggregated and the coating has the lowest hardness. A large number of hard particles are peeled off and are carried away from the frictional interface under the action of seawater erosion, which makes it difficult to form a lubricating film. That is the reason for severe wear on T2.5. To further clarify the effect of seawater corrosion on the tribological properties of the coating, the corrosion resistance of the coating was tested. Coated samples with different Ce_2_O_3_ contents were immersed in the same amount of seawater and their surfaces were photographed every 24 h.

The wear morphology of different coatings under seawater corrosion conditions is shown in Figure 11. The figure shows a large number of furrows on the surfaces of both T1.5 and T2 coatings, and the wear marks are smooth compared with those under dry sliding wear. The scouring effect by the water is the reason. The wear mechanism is abrasive wear, where the T2.5 coating is worn through and exposed to the substrate. The wear and tear on the coatings are much more severe under seawater corrosion than that under dry friction. The reasons are listed as follows. As the samples are working in corrosive seawater, the wear of the coating is affected by both corrosion and sliding wear. The cracks and separations caused by sliding wear cause corrosive medias to enter the coating. The coating’s resistance to wear is reduced under the corrosion. On the other hand, a large number of particles are peeled off by the high-frequency reciprocating movement. This leads to a deterioration of the wear in the case of seawater corrosion. However, seawater can also form a film of water, which has the function of lubrication and can carry away much of the frictional heat generated in the frictional process. At the same time, seawater erosion also aggravates the loss of the coating particles, resulting in worse wear. As Figure 5 shows, with the addition of Ce_2_O_3_, the hardness of coatings gradually decreases, the ability of coatings to resist abrasive wear decreases. Thus, the phenomenon of wear aggravation occurs.

To further amplify the corrosion effect, the sample was immersed for an extended time. Figure 12a–c show the surface corrosion of different coatings. As the Ce_2_O_3_ content rises, the bubbles on the surface of the coating gradually decrease, and the T2.5 coating has the lowest number of bubbles. The reason is that the inclusion of Ce_2_O_3_ particles in the T2.5 coating is the largest, and Ce^3+^ is present in Ce_2_O_3_ particles. The study of Fan et al. [37] showed that Ce^3+^ was easy to convert into Ce^4+^, which hindered the charge transfer and thus prevented the corrosion reaction. The study of Zhang et al. [45] showed that, with the progress of the corrosion reaction, the corrosion products of Ce^3+^ become stable Ce(OH)_3_ and Ce(OH)_4_, which can fill in the microcracks and act as a barrier to the penetration of a corrosive medium. They prevent the corrosion from further penetrating the coating, thus improving the coating’s corrosion resistance. The T2.5 coating has the highest amount of Ce_2_O_3_ particles and has the best corrosion resistance. However, there are different results that are presented in Figure 12d–f. The poorest corrosion and many bubbles grew around the wear mark of the T2.5 coating. The number of bubbles on the T1.5 coating was the least among these coatings. The main reason is the coating thickness on the wear mark. The coating with 2.5 wt% is worn out, and the coating with 1.5 wt% has the thickest coating on its wear mark. The anti-corrosion ability increases with the coating thickness. Thus, the corrosion of the coating increases with the content of Ce_2_O_3_ due to the influence of the wear mark.

Taking the different coatings under different working conditions into consideration, it is found that the coating with a Ce_2_O_3_ content of 2 wt% has the best tribological performances under dry sliding wear and oil lubrication. The average CoF of a coating with a Ce_2_O_3_ content of 2 wt% is 0.128 under seawater, and the CoF of a coating with 1.5 wt% Ce_2_O_3_ is 0.124. As the Ce_2_O_3_ increases from 1.5 wt% to 2 wt%, the CoF increases by just 3.23%. Thus, the coating under 2 wt% Ce_2_O_3_ can be regarded as the best material. The hardness and elastic modulus of the coating under 2 wt% Ce_2_O_3_ is the biggest among these different coatings. The appropriate amount of Ce_2_O_3_ particles improves the anti-wear and anti-corrosion performances under different working conditions.

## 4. Conclusions

(1)The coating with 2 wt% Ce_2_O_3_ has the best wear resistance under dry sliding wear. As the content of Ce_2_O_3_ is lower than 2 wt%, the wear mechanisms of these coatings are abrasive wear. As the content of Ce_2_O_3_ increases to 2.5%, it changes into adhesive wear. Since the agglomeration particles increase with the content of Ce_2_O_3,_ the coating with 2.5 wt% Ce_2_O_3_ has the poorest tribological properties;(2)The CoFs of coatings under seawater have the biggest fluctuations. The Ce_2_O_3_ particles are washed out by the scouring effect of high-frequency reciprocating frictional experiments. Though the anti-corrosion is increased with the content of Ce_2_O_3_, the coating with 2.5 wt% Ce_2_O_3_ has the poorest tribological and corrosive performances;(3)The oil film on the surface of the frictional pair has a lubrication and protection effect on the coating. Compared with dry sliding wear and seawater, the frictional coefficient and wear rate of coatings under oil lubrication have the best performance.

## Figures and Tables

**Figure 1 polymers-15-01507-f001:**
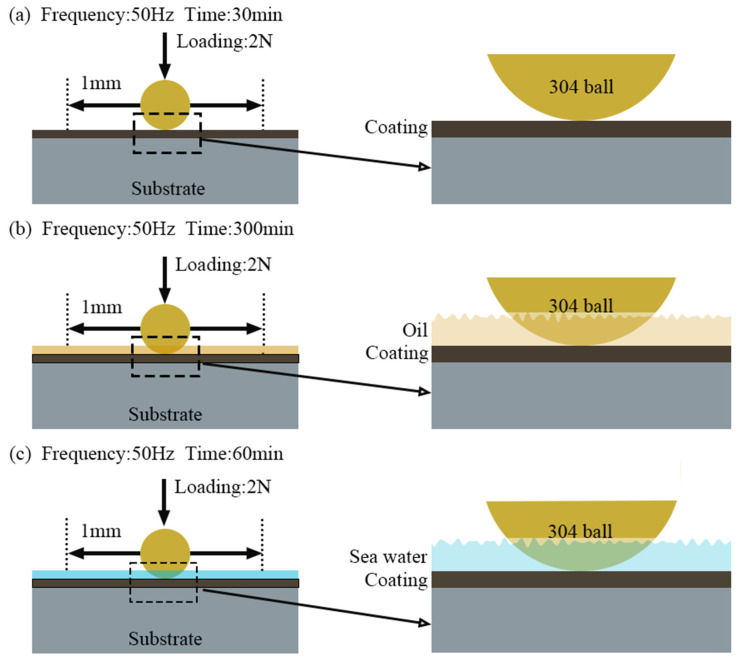
Test conditions of tribological experiments. (**a**) dry sliding wear; (**b**) oil lubrication; (**c**) seawater corrosion.

**Figure 2 polymers-15-01507-f002:**
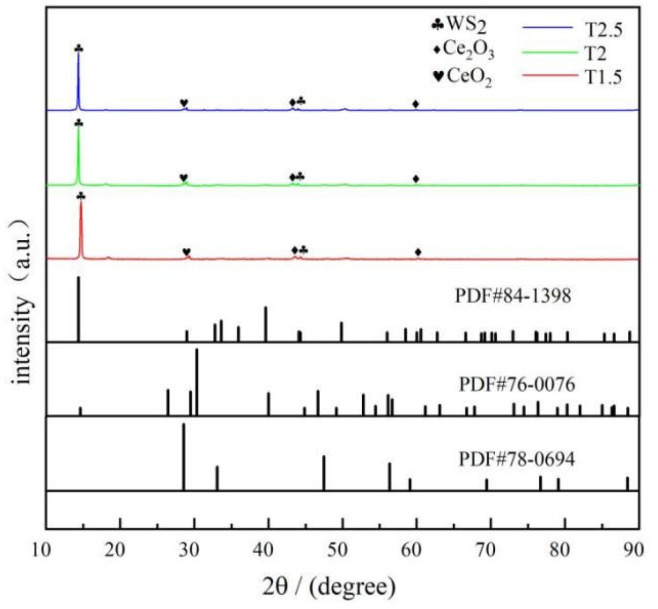
XRD pattern of composite coatings.

**Figure 3 polymers-15-01507-f003:**
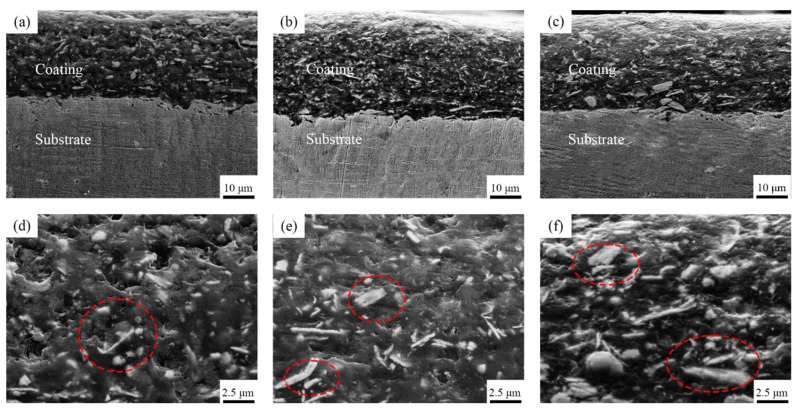
Cross-sectional micromorphology of composite coatings. (**a**) T1.5 coating; (**b**) T2 coating; (**c**) T2.5 coating; (**d**) Enlarged image of (**a**); (**e**) Enlarged image of (**b**); (**f**) Enlarged image of (**c**).

**Figure 4 polymers-15-01507-f004:**
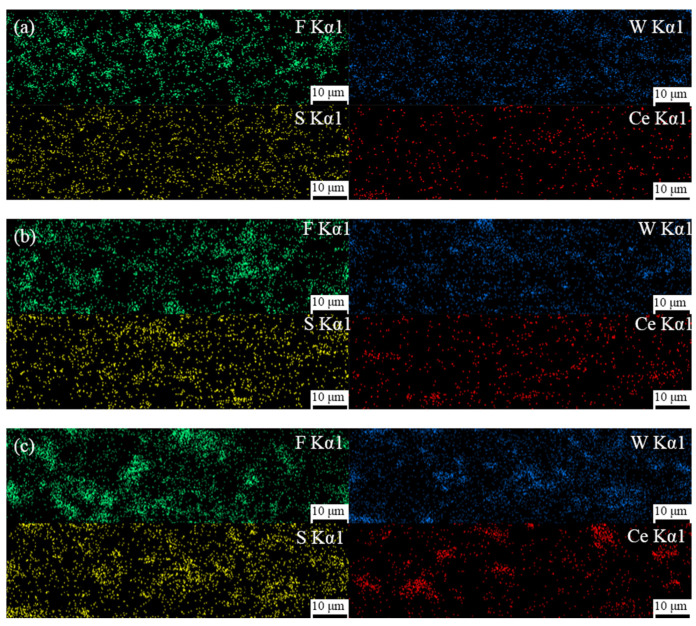
EDS scan results of composite coatings. (**a**) T1.5 coating; (**b**) T2 coating; (**c**) T2.5 coating.

**Figure 5 polymers-15-01507-f005:**
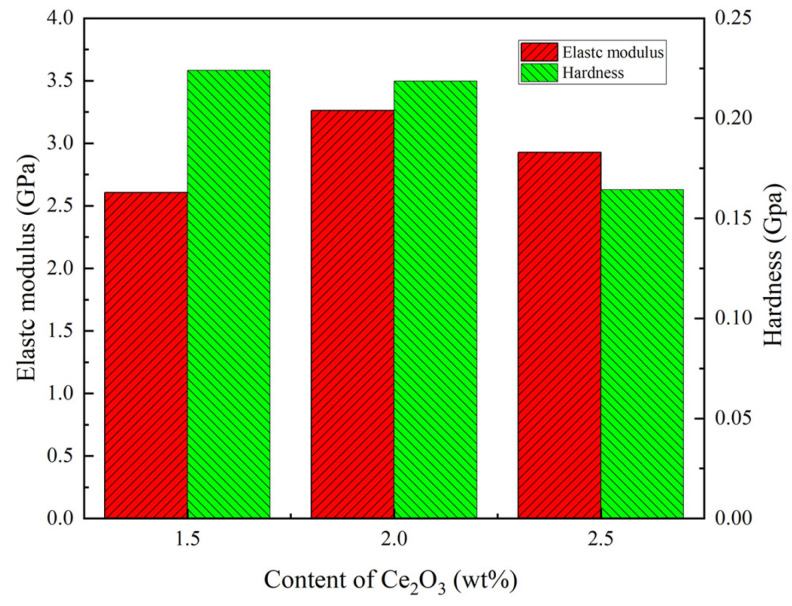
Hardness and elastic modulus of coating.

**Figure 6 polymers-15-01507-f006:**
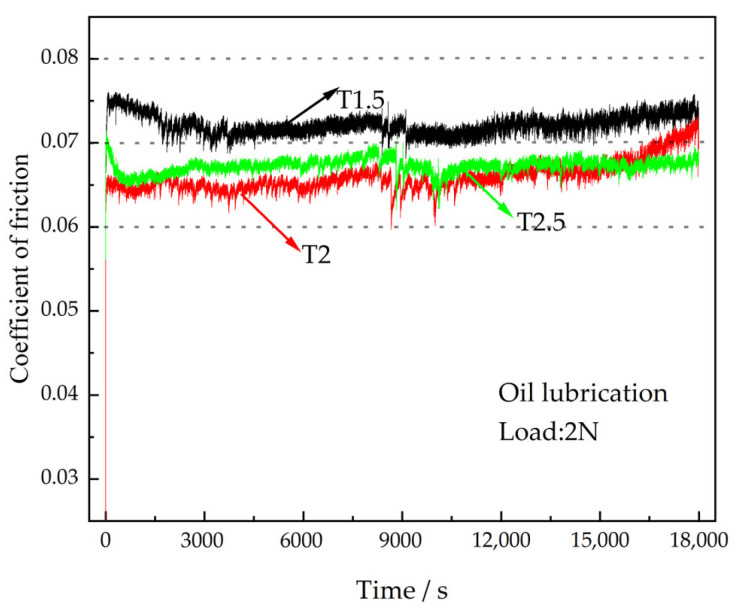
CoFs of the coatings under oil lubrication.

**Figure 7 polymers-15-01507-f007:**
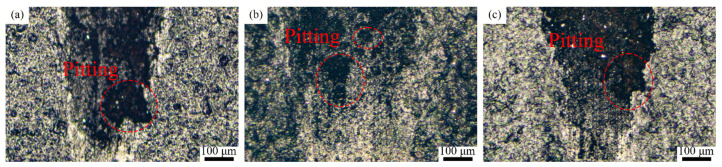
Wear marks of the composite coating under oil lubrication conditions. (**a**) T1.5 coating; (**b**) T2 coating; (**c**) T2.5 coating.

**Figure 8 polymers-15-01507-f008:**
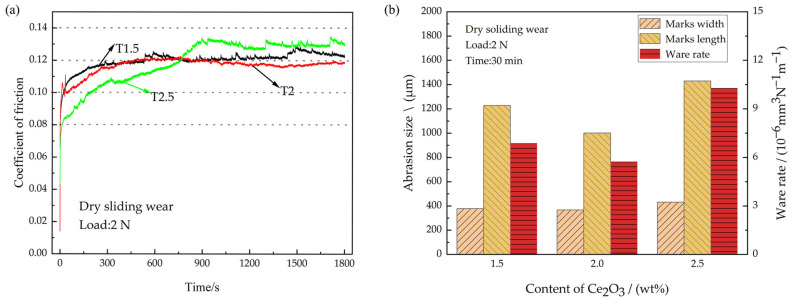
CoFs and wear amount under dry sliding wear. (**a**) Coefficient of friction; (**b**) Wear amount.

**Figure 9 polymers-15-01507-f009:**
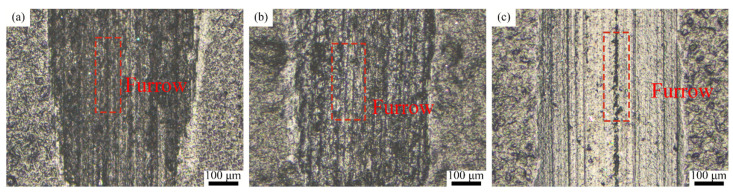
Wear marks of the composite coating under dry sliding wear. (**a**) T1.5 coating; (**b**) T2 coating; (**c**) T2.5 coating.

**Figure 10 polymers-15-01507-f010:**
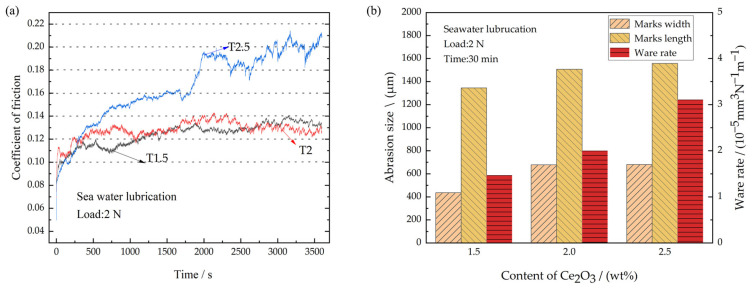
Friction coefficient and wear amount under seawater corrosion. (**a**) Coefficient of friction; (**b**) Wear amount.

**Figure 11 polymers-15-01507-f011:**
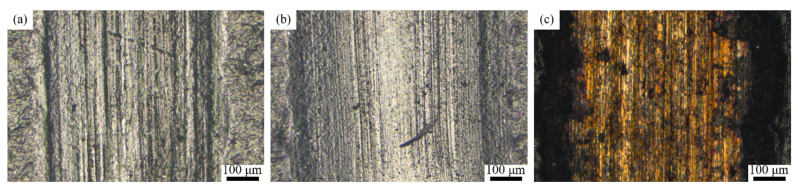
Wear marks of the composite coating under seawater corrosion. (**a**) T1.5 coating; (**b**) T2 coating; (**c**) T2.5 coating.

**Figure 12 polymers-15-01507-f012:**
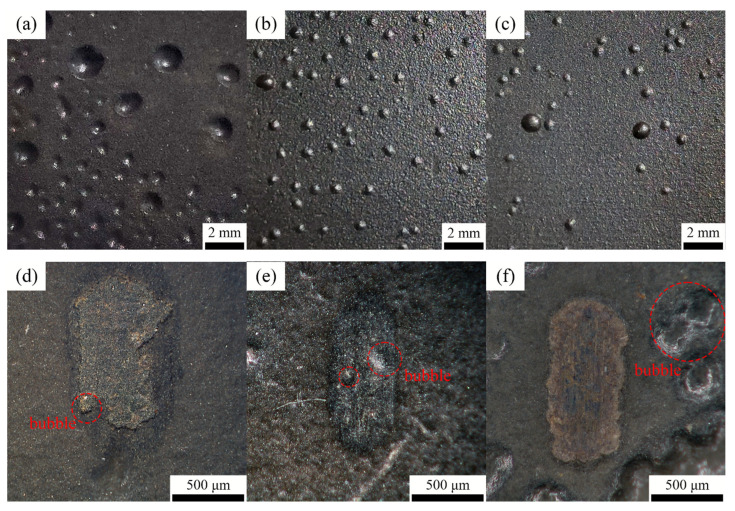
The surface of the coating after 18 days immersed in seawater. (**a**) T1.5 coating; (**b**) T2 coating; (**c**) T2.5 coating; (**d**) T1.5 wear marks; (**e**) T2 wear marks; (**f**) T2.5 wear marks.

**Table 1 polymers-15-01507-t001:** Compositions and contents of three coatings.

Samples	Solvent (A)	Binder (B) PI+PAI+Ep-44 (wt%)/5:3:4.5	Lubrication (C)		Additive (D) (wt%)	Filler (E) Ce_2_O_3_ (wt%)
			PTFE (wt%)	WS_2_ (wt%)		
T1.5	70	12.5	4.5	4	7.5	1.5
T2	70	12.5	4	4	7.5	2
T2.5	70	12.5	3.5	4	7.5	2.5

**Table 2 polymers-15-01507-t002:** Chemical composition and related parameters of the frictional pairs.

Frictional Pair	Chemical Compositions (wt%)												
	C	Mn	P	S	Cr	Ni	Cu	Sb	Al	Fe	Pb	Sn	Zn
CuPb22.5Sn2.5	-	-	0.1	-	-	0.5	70.2	0.5	-	0.7	22.5	2.5	3
Stainless steel ball	0.08	2	0.045	0.03	20	11	-	-	66.8	-	-	-	0

## Data Availability

The data underlying this paper cannot be shared publicly due to the privacy concerns of the individuals involved in the study.

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
