# Peer review of "The Tribological and Mechanical Properties of PI/PAI/EP Polymer Coating under Oil Lubrication, Seawater Corrosion and Dry Sliding Wear"

_polymers, 2023, doi:10.3390/polym15061507_

Round 1

Reviewer 1 Report

The abstract is too long. Basic guidelines should be given: what is the purpose of the development; what methods are used; and what is the main conclusion of the development?

The abstract should be a total of about 200 words maximum. 

The article is not written according to the requirements of the journal.

Point 2 should be revised as required.

The conclusion is unrelated to the title.

The main conclusions must be rethought and rewritten.

Reviewer 2 Report

• How do the authors select a particular coating concentration of Ce2O3? What can be the effect if a high or low concentration is selected?

• Whether the stainless steel ball with a diameter of 6 mm can replicate the sea-water condition tests for parts manufactured for the high impact pressure?

• Hertzian contact stress was 2 N. Is it the same for all conditions? Is there any effect of particle concentration on contact stress?

• The author can show the Acoustic response of friction based on the reciprocating test.

• Write the chemical composition of the stainless steel ball.

• It is better to correlate the friction and wear performance of a developed coating in relation to dry test, oil test, and sea-water condition. A detailed mechanism on the interfaces can better explain the physics involved.

Round 2

Reviewer 2 Report

Accept in present form